# The Impact of a Commercial Biostimulant on the Grape Mycobiota of *Vitis vinifera* cv. Barbera

**DOI:** 10.3390/microorganisms11081873

**Published:** 2023-07-25

**Authors:** Laura Pulcini, Elisa Bona, Enrico Tommaso Vaudano, Christos Tsolakis, Emilia Garcia-Moruno, Antonella Costantini, Elisa Gamalero

**Affiliations:** 1Consiglio per la Ricerca e l’analisi dell’Economia Agraria—Centro di Ricerca Viticoltura ed Enologia (CREA-VE), Via P. Micca 35, 14100 Asti, Italy; enricotommaso.vaudano@crea.gov.it (E.T.V.); christos.tsolakis@crea.gov.it (C.T.); emilia.garciamoruno@crea.gov.it (E.G.-M.); 2Dipartimento di Scienze e Innovazione Tecnologica (DISIT), Università del Piemonte Orientale, Viale T. Michel 11, 15121 Alessandria, Italy; elisa.gamalero@uniupo.it; 3Dipartimento per lo Sviluppo Sostenibile e la Transizione Ecologica (DISSTE), Università del Piemonte Orientale, Piazza San Eusebio 5, 13100 Vercelli, Italy; elisa.bona@uniupo.it

**Keywords:** grape, mycobiota, yeast, biodiversity, Acibenzolar-S-Methyl, next-generation sequencing, ITS region

## Abstract

Reducing the use of fungicides, insecticides, and herbicides in order to limit environmental pollution and health risks for agricultural operators and consumers is one of the goals of European regulations. In fact, the European Commission developed a package of measures (the European Green Deal) to promote the sustainable use of natural resources and strengthen the resilience of European agri-food systems. As a consequence, new plant protection products, such as biostimulants, have been proposed as alternatives to agrochemicals. Their application in agroecosystems could potentially open new scenarios regarding the microbiota. In particular, the vineyard microbiota and the microbiota on the grape surface can be affected by biostimulants and lead to different wine features. The aim of this work was to assess the occurrence of a possible variation in the mycobiota due to the biostimulant application. Therefore, our attention has been focused on the yeast community of grape bunches from vines subjected to the phytostimulant BION^®^50WG treatment. This work was carried out in the CREA-VE experimental vineyard of *Vitis vinifera* cv. Barbera in Asti (Piedmont, Italy). The composition of fungal communities on grapes from three experimental conditions such as IPM (integrated pest management), IPM+BION^®^50WG, and IPM+water foliar nebulization was compared by a metabarcoding approach. Our results revealed the magnitude of alpha and beta diversity, and the microbial biodiversity index and specific fungal signatures were highlighted by comparing the abundance of yeast and filamentous fungi in IPM and BION^®^50WG treatments. No significant differences in the mycobiota of grapevines subjected to the three treatments were detected.

## 1. Introduction

Studies on microbiota are currently widespread in many scientific fields [1]. Since the publication of the initial papers on human microbiota, the scientific community’s interest in this topic has extended to other environments, including food production chains [2,3]. The wine sector is particularly significant in the Italian economy, with wine playing a crucial role in the export of Italian agri-food. Despite the challenges posed by the COVID-19 pandemic, this remains true even in recent years [4,5,6,7,8]. In fact, in 2021, Italy exported EUR 7113 million of wine, which represented 13.7% of exported agri-food products [9]. In Piedmont, one of Italy’s largest wine-growing regions, the trend of vineyard cultivation is similar to the Italian one [10].

Due to the economic impact of this sector, one of the main targets of viticulture and enology research is to reach a complete knowledge of the role that microorganisms can have on grapevine health, productivity, and wine quality throughout the entire production chain [1,11,12,13,14,15]. For many years, scientists have tried to explain the effects of microorganisms on wine features. Based on the idea that the typical sensory characteristics of wine would be related to the composition of the microbial community living on grapes and in must [16,17,18], a large number of papers focused on the identification and selection of culturable wine yeast and lactic acid bacteria able to carry out alcoholic and malolactic fermentation [19,20]. The strict relationship between the “terroir” (the wine’s sensorial characteristics as defined by the cultivation environment, vine variety, vineyard management, and enological techniques applied) and the microbiota occurring in different geographical regions has been demonstrated [17,21,22,23,24,25]. Moreover, the vine is subject to a series of biotic adversities requiring different management choices. As far as arthropods are concerned, the ones that cause the most damage to the plant and production are the grapevine moth *(Lobesia botrana* [26] and *Eupoecillia ambiguella*) and the grapevine yellow spider-mite *(Eotetranychus carpini* f. *vitis*). Fungal adversities include downy mildew (*Plasmopara viticola*), powdery mildew (*Oidium tuckeri*), and grey mold (*Botritis cinerea*) affecting grapevine yield. Fungi such as *Phaeoacremonium aleophilum*, *Phaeomoniella chlamydospore,* and *Fomitiporia mediterranea* simultaneously colonize the plant and can impair the physiology of the vine, leading to Esca Disease [27]. Fungal diseases could develop from insect-feeding areas. Bacteriosis (e.g., *Agrobacterium tumefaciens*), virosis (e.g., grapevine leafroll associated virus—GLRaV, grapevine rupestris stem pitting associated virus—GRSPaV) and phytoplasma disease (grapevine flavescence dorée 16SrV) are important ampelopathies for which good prophylaxis and phytoiatric strategies are necessary [28]. Quarantine organisms also exist for vines (e.g., “*Candidatus* Phytoplasma vitis”) and are listed in Regulation (EU) 2019/2072 (Annex II).

In the past, ampelopathies have been cured by synthetic biocide molecules belonging to dithiocarbamates, organophosphorus, sterol biosynthesis inhibitors (SBIs), strobilurins, copper-based or sulfur fungicides, etc. The massive use of these molecules led to the selection of resistant pathogens and the pollution of the biosphere with the loss of the biological fertility of soils [29]. In fact, the new dispositions of the European Law (Green Deal, COM (2019) 640 final and ANNEX) relate to the reduction in agropharmaceuticals and the withdrawal of many fungicides. 

The prevalent strategy to preserve the physiology of the vine, the productivity, and the grape quality is represented by the use of biostimulants as resistance inducers or elicitors (chitosan, nitrogen biostimulant, amino acids, etc.), which are formulated for foliar applications [30,31,32].

Biostimulants are commonly used against a broad spectrum of microorganisms causing disease in several fruit trees and crops. Their mechanisms of action take advantage of the endogenous and systemic circulation of pathogen-associated molecular patterns (PAMPs) signaling cascade that induces an immune response throughout the whole plant. Therefore, biostimulants stimulate the endogenous salicylic acid (SA) production or/and activate Systemic Acquired Resistance (SAR). The expression of SAR genes leads to the synthesis of different families of molecules named pathogenesis-related proteins (PR) such as chitinase, glucanase, endoprotease, and peroxidase, and all are involved in soil-borne disease suppression. One of the synthetic molecules with high effectiveness to prevent bacteriosis is the propesticide Acibenzolar-S-Methyl (ASM) in the benzothiazoles chemical category, which can mimic SA behavior [33,34,35,36,37]. An agrochemical product by Syngenta, containing 50% of the weight of ASM, is commercialized as BION^®^50WG in Europe. It has both phloematic and xylematic circulation and is effective against many plants’ bacterial diseases. According to the label of the commercial product, BION^®^50WG is effective against bacterial diseases of some fruit plants and the downy mildew of tobacco (*Peronospora tabacina*). Moreover, ASM is known as BTH (benzo (1,2,3) thiadia-zole-7-carbothioic acid S-methyl ester) and was used as a fungicide on a variety of crops; for example, in the control of the powdery mildew of wheat and barley [37,38]. ASM is classified as a plant activator, antifungal agrochemical, and profungicide (the corresponding carboxylic acid is released by hydrolysis of the thioester group) [38]. It is reasonable to think that this biostimulant could affect the composition of the grapes’ microflora with consequences on must and, subsequently, wine quality. At the moment, in Italy, BION^®^50WG is not authorized for application on vines. However, in 2021, the Ministry of Health temporarily authorized this compound on vines for 120 days in order to hinder “*Candidatus* Phytoplasma vitis”, an unresolved threat (DD 19 May 2021 Ministero salute [39]). Considering the possible definitive approval of this molecule on grapevine in the near future, we decided to set up a trial carried on in the CREA-VE experimental vineyard in order to investigate the possible effect of BION^®^50WG on the community of yeasts living on the grape berries by using a metabarcoding approach [13,40,41,42].

The next-generation sequencing (NGS) technique is the most effective to study the environmental microbiota in a few steps by obtaining a considerable number of Operational Taxonomic Units (OTUs) and data on the basis of metagenomic elaborations. To the best of our knowledge, the existing literature on the biostimulant’s effect on grape microbiota is poor and deserves to be improved [43,44]. 

## 2. Materials and Methods

### 2.1. Field Trial: Experimental Setup and Plant Treatments

The vineyard is in the countryside close to Asti (latitude: 44°55′18.7″ N.; longitude: 8°11′44.0″ E) at an altitude from 166 to 198 m a.s.l. and a slope of about 17%. It faces south and has rows arranged in an east–west orientation. Piedmont is characterized by a continental climate. In 2019, it had an average minimum temperature of 1.92 °C and a maximum temperature of 25.54 °C, and 803.4 mm of annual rainfall was recorded (personal communication from Agenzia Regionale per la Protezione Ambientale (ARPA), Piemonte). The vineyard is extended on 2 ha and is composed of 63 rows cultivated with 6 *V. vinifera* varieties, which were planted in 2006. The grapevine area considered for the analysis (0.5 ha) is highlighted in white in Figure 1B. All the plants were grafted on 1103 Paulsen rootstock (high vigor; adapts well to clayey, compact, and saline soils; absorbs a lot of magnesium and postpones the ripening). The grapevine of about 1 m height is cultivated in the Guyot training system based on leaving a spur and a new shoot for future fruiting with the pruning technique; in this way, after pruning, 6–12 buds and 2 buds occur on the shoot and spur, respectively.

The vineyard was subjected to integrated crop management (ICM) both for cultivation techniques and biotic threats (fungal and bacterial pathogens and insects). As described in the FAO textbook “Policy Support Guidelines for the Promotion of Sustainable Production Intensification and Ecosystem Services”, the integrated crop management strategies consist of the cultivation through the implementation of every possible action to reduce the use of synthetic chemicals in agriculture and application of the good agricultural practices. This guideline promotes (i) no tillage where suitable, (ii) maintaining the soil cover with residues, (iii) crop rotation or association, (iv) balanced nutrition (no excessive nitrogen fertilization), (v) integrated pest management (IPM) using selective active molecules with a minimum impact on the environment and human health, and (vi) damaging species (arthropods and cryptogams) need to be managed when a certain infestation threshold is reached [45,46].

Therefore, both fungicide and insecticide treatments were performed according to the phytosanitary bulletin of the Piedmont Region.

The experimental trials were carried out during the 2018 and 2019 vegetative seasons according to the experimental protocol organized in the project “*Elicitori di resistenza a supporto della difesa dalla Flavescenza dorata della vite*” (“Resistance elicitors to support vine defense against the Flavescence dorée”) coordinated by Istituto per la protezione Sostenibile delle Piante of the Consiglio Nazionale delle Ricerche (CNR-IPSP) and funded by the Regione Piemonte [47,48]. The plant stimulant BION^®^50WG by Syngenta (Basel, Switzerland) (Acibenzolar-S-methyl 50% in hydro dispersible granules) was applied. 

In this work, three treatments were considered, which were referred to as (Figure 1):
-Control (C): Integrated pest management (IPM) of the vineyard comprising the use of insecticides and fungicides against the main parasites and molds by considering their infestation in the vegetative stage, especially during ripening time.-Bion (B): IPM combined with the plant biostimulant BION^®^50WG (Syngenta). It was applied every 2 weeks in 2018 (on 26 June, 10 July, 24 July, 7 August, 21 August) and 2019 (26 June, 10 July, 24 July, 7 August) by leaf irroration with 500 L of water suspension. Considering the similarity in plant habitus, the treatment protocol was inspired by the posology recommended for the kiwi plant.-Water (W): Integrated pest management combined with the nebulization of 500 L of water on leaves to simulate the possible effect of the humidity increase without the active molecule.

Soil sampling was performed at ripening time. The soil was sampled (N = 7; three replicates) at a depth of 30 cm after removing the surface layer. Physical/chemical analyses were performed on each soil sample according to D.M. 13 September 1999.

### 2.2. Grape Sampling

Grape sampling was performed on 3 October 2019, the day before the beginning of the harvest. According to the literature [49], this is the best moment to collect indigenous fermentative yeasts and evaluate the microbiological conditions of the berries.

The sampling area consists of 22 rows of grapevine cv. Barbera (from row 23 to row 44). Sampling started from the top of the vineyard. Both rows 1 and 22, representing the upper and the lower rows in terms of altitude, were excluded to avoid the extreme positions. Row 12 was excluded because plants were treated as the controls with the aim to create a buffer area between the Bion and Water and avoid the drift phenomenon (Figure 1C,D). Grapevines showing flavescence dorée and Esca Disease symptoms [48], as well as those showing symptoms related to aridity and soil erosion (stunted vegetation or immature grapes), were excluded by the grape sampling. In order to ensure a randomized sampling within the theses, samples were collected according to the following criteria: (i) 5 vines far enough away from each other have been considered for each thesis (Figure 1D); (ii) 2 bunches (total weight of about 700 g) were detached from each grapevine and collected in sterile bags [50]; and (iii) the collected grapes were stored in a cooler bag within frozen tablets in order to preserve the berry integrity (10 °C). Each sample was processed to investigate the mycobiota of the berries’ skin [51,52].

### 2.3. Yeast Collection and Storage

In order to collect the yeasts from grape skin, 50 berries (average weight of the berries 2.60 ± 0.24 g) were randomly sampled from each bunch using sterilized scissors, avoiding any juice release. Berries were put into a sterile flask with 200 mL of sterile saline solution (0.9% NaCl) and incubated at 25 °C overnight under shaking at 75 rpm [49]. Fifteen mL of suspensions were pelleted by centrifugation at 3000× *g* for 15 min at 4 °C. The pellets were resuspended in 5 mL of sterile saline solution and these cellular suspensions, consisting of microorganisms and material surrounding grape berries, were used for the metabarcoding analysis.

### 2.4. Mycobiota Characterization

Genomic analysis of the surface berry mycobiota was carried out on cells pelleted as described above. DNA extraction and purification were performed by a handmade CTAB method starting from a pellet of 1 mg [41]. The amount of DNA was assessed by a spectrophotometer reading at 260 nm (Beckman Coulter DU700, Brea, CA, USA), and purity was evaluated at a 260/280 ratio.

The primers were carefully chosen by taking into account that the ITS region of the fungi could overlap with the same region of the higher plants. Thanks to the accurate work of sampling on bunches with scissors, we avoided the damage of berries and consequent leakage of vine nucleic acid. Therefore, the plant genome did not affect the NGS analysis and the last one was focused exclusively on the whole genome of the fungi community resident on the berries’ surface. Primers ITS1F (5′-TCCGTAGGTGAACCTGCGG-3′) and ITS4R (5′-TCCTCCGCTTATTGATATGC-3′) were chosen for their high specificity and coverage of the ITS region [53,54]. As recommended by the Illumina protocol, a PCR reaction with a hot start high-fidelity DNA polymerase (Roche, Monza, Italy) and the ITS1F and ITS4R primers was performed to verify the quality of the purified DNA [53,54,55]. The ITS DNA libraries (hypervariable regions ITS1-5.8S-ITS2) were obtained by two amplification steps. An initial PCR amplification using locus-specific PCR primers (detailed above) and a subsequent amplification that integrates relevant flow–cell binding domains and unique indices (NexteraXT Index it. Illumina Inc., San Diego, CA, USA) were performed. The libraries were sequenced using the 300 bp paired-end mode and, the processing of the amplicon pool was processed on an MiSeq instrument (Illumina, San Diego, CA, USA).

### 2.5. Bioinformatic and Statistical Analysis

The obtained reads were de-multiplexed based on the Illumina indexing system. Where the amplicon length was permissive with the respective sequencing length, 3′ ends of the pairs overlapped to generate consensus pseudo-reads, while the remainder were maintained as separated pairs. Then, a clipping routine was applied to remove low-quality bases at 3′ tails. Furthermore, any primer sequence at 5′ ends was removed and not accounted for during the process. All reads were used in the analysis if they maintained a minimum length of 200 bp after the removal of primer sequences and low-quality bases. Paired reads with permissive overlap at their 3′ ends were merged into a single fragment and used to improve assignment accuracy.

Following the QIIME pipelines, the USEARCH algorithm (version 8.1.1756. 32-bit) allowed the steps of chimera filtering, grouping of replicate sequences, sorting sequences according to decreasing abundance, and Operational Taxonomic Unit (OTU) identification.

OTU picking aims to group query sequences into clusters represented by centroids.

The query sequences not sharing similarity with a centroid were discarded. 

Each centroid shares a level of similarity with its member sequences. These fragments (sequences clustered in centroid) were aligned to a specific reference database for the molecular identification of fungi, UNITE2016 [56]. This database offers 1,000,000 public fungal ITS (nuclear ribosomal internal transcribed spacer region) sequences for reference by establishing the taxonomic affiliation. Only matches with a minimum identity of 94% were retained and subjected to further classification. The database sequences were maintained as representative sequences of OTUs.

The Ribosomal Database Project (RDP) classifier and reference database were used to assign taxonomy with a minimum confidence threshold of 0.50.

The total count is retained for alpha and beta diversity estimators, taxonomic abundance estimation, and ad-hoc statistical tests.

After this analysis, data in BIOM format were loaded on Microbiome-Analystwebsite with R version 4.1.3, where consolidated taxon set libraries were available with the “Marker Data Profiling—MDP” tool [57,58]. Before data analysis, a data integrity check was performed on the platform to ensure that all the information necessary was collected. Then, data filtering was performed in order to improve the results and identify and remove features that are unlikely to be useful when modeling the data. In particular, during the filtration step, features with a low count and variance can be removed, while those with very few counts can be filtered based on their abundance levels (minimum counts 10) across samples (prevalence). To bring the samples at the same scale for comparison, a three-step normalization procedure (consisting of data rarefaction, data scaling, and data transformation) was performed. Rarefaction and scaling methods allowed us to deal with the uneven sequencing depths by obtaining samples at the same scale for comparison [59].

Alpha diversity was extrapolated from Microbiome-Analyst. The statistical significance of grouping based on experimental factors was estimated using the Kruskal–Wallis non-parametric test. The species richness (the total number of observed species) and the beta diversity measure the similarity in terms of the abundance of yeast-sampled species among the samples. Both alpha and beta diversity were obtained using the Phyloseq package [58,60]. The Shannon (H) and Simpson (D) indexes, which consider the abundance of organisms (evenness), were used to describe the diversity of the fungal community. The results were plotted across samples and reviewed as box plots for each considered group (sampling site).

Beta diversity was obtained by measuring the distance (dissimilarity) among the samples. Each sample was compared to every other sample, generating a distance matrix. Using Bray–Curtis distance and Principal Coordinate Analysis (PCoA), it was possible to visualize these matrices in a 2D plot, where each point represents the entire microbiome of a single sample. Each axis reflects the percentage of variation between the samples, with the x-axis representing the highest dimension of variation and the y-axis representing the second highest dimension of variation. Each point or sample displayed on PcoA plots is colored based on either sample group (treatment). Moreover, the statistical significance of the clustering pattern in the ordination plots can be evaluated using permutational ANOVA (PERMANOVA).

A hierarchical cluster analysis was performed at the species level, where each sample begins as a separate cluster and the algorithm proceeds to combine them until all samples belong to one cluster. Two parameters were considered: (i) the distance measured between the samples (Bray–Curtis distance) and (ii) the clustering algorithms, including average linkage results shown as a heatmap (distance measure using Euclidean and clustering algorithms using ward.D at the species level).

The heat tree method was used to compare abundance at the species level. Heat trees use a hierarchical structure of taxonomic classifications to depict quantitatively and statistically the taxon differences among fungi communities. The quantitative data refer to the median abundance and the non-parameter statistical test Wilcoxon ranksum, which was applied by comparing one pair of the theses at a time. The resulting differential heat tree indicates which species are more abundant in the different considered treatments. A heat tree analysis was performed using the metacoder R package, according to Foster [61].

A core microbiome analysis [62] was performed in order to identify core species on the berries’ skin in the different theses that remained unchanged in their composition across the whole fungal community. A core microbiome analysis is adopted from the core function in the microbiome R package. The result of this analysis is represented in the form of a heatmap of core taxa where the y-axis represents the prevalence level of core features and x-axis represents the detection threshold range (relative abundance).

Moreover, a PCA analysis was performed using all the considered parameters based on the “sampling site” factor by using R (v. 3.5.1) (R Core Team 2018) [63]; in particular, the FactoMineR [64] and Factorextra [65] packages.

Finally, in order to identify the signature associated with the different parameters, the Linear Discriminant Analysis Effect Size (LDA-LefSe) method was applied at the species level. This method is specifically designed for biomarker discovery and explanation in high-dimensional metagenomic data [66]. It incorporates statistical significance with biological consistency (effect size) estimation. It performs a non-parametric factorial Kruskal–Wallis (KW) sum-rank test to identify species with significant differential abundance with regard to the factor of interest (treatment) followed by a linear discriminant analysis (LDA) to calculate the effect size of each differentially abundant feature. The result consists of all the species with the highest mean and the logarithmic LDA score (effect size). The features are significant based on their adjusted *p*-value. The default *p*-value cut-off was 0.05.

## 3. Results

### 3.1. Soil Analysis

Cation exchange capacity (CEC) is a soil chemical property that measures a soil’s ability to hold nutrients. So, it is a key determinant of soil fertility. The soil analyzed in this work was clayey loam with a cation exchange capacity (CEC) of 27.91 meq/100 g, which denotes quite good soil fertility (Table 1) [67]. 

### 3.2. Mycobiota Characterization

A total of 3,969,382 reads were obtained with a mean value of 264,625 reads per sample. After the demultiplexing step, a total of 1,737,580 reads (with a mean value of 115,839 reads per sample) were used for further analyses. The genomic sequences were included in the BioProject PRJNA801453 “Study of the yeast microbiota of grape skin” available in the NCBI database accessed on 28 January 2022. The BioProject contains 15 BioSamples with accession IDs from SAMN25351049 to SAMN25351063. In the 15 samples, 1882 features (taxa) were present. After the data filtering step, 1772 features remained. 

The rarefaction curves (Figure 2), resulting from the statistical analysis of the OTUs, showed that in each sample a huge number of fungal OTU were sequenced (minimum 197 in R8P1C, maximum 403 in R10P16B). Good sequencing coverage of all the samples processed was obtained, which was always greater than 99.5% [68]. 

The evaluation of three alpha diversity estimators was accomplished at the species level according to the theses (Figure 3). The number of observed species on the grape skin differed significantly according to the treatment (*p* = 0.04999) (Figure 3A, Appendix A). Shannon’s diversity index did not differ according to the treatment (*p* = 0.73345) (Figure 3B). Simpson’s index (Figure 3C) showed no differences between the theses (*p*= 0.56553). Thus, there is a high probability that the two OTUs chosen at random are the same, which is reflected in the high D of about 0.65 for all samples.

Beta diversity (the comparison of fungal communities based on their composition) provides a measure of the distance or dissimilarity between each sample pair. Principal Coordinates Analysis (PCoA), performed on the observed species (Figure 4), indicates that axis 1 explains 73.4% of the diversity and the second one explains 12.1%. The overall composition of the grape skin mycobiota, considered at the species level, was not significantly affected by the sampling site (treatment), as indicated by non-parametric multivariate analysis of variance testing (PERMANOVA; *p* < 0.001).

Since our data indicate that the sampling site (different treatments properties) did not affect the biodiversity of the grape fungal communities in the 15 grapevines considered, we decided to construct a core microbiome analysis at the species level to represent the main yeast living on the berry surface (Figure 5). *Hanseniaspora uvarum* represented the dominant species in each thesis. *Issatchenkia terricola* was detected in the Bion and Water, appearing prevalent in this last treatment, and was not represented in the Control treatment. *Candida californica* appears in the core mycobiota exclusively in the water treatment. A relevant amount of the reads was assigned to *Incertae sedis* at the species level but from the core microbiota analysis at the order level, the predominant outcomes were the *Saccharomycetales*. Consequently, most parts of the uncertain assignments must be assigned to this order.

Considering the fungal relative abundance at the species level, in each case, a relevant percentage of *Incertae sedis* was detected, but these fungi were classified at higher taxonomic levels. The reads showing not assigned did not match with any OTUs in the reference database.

A graphical view using the heat tree method of the abundance at the species level is shown in Appendix A.

The heatmap representing the abundance of the different fungal species (Figure 6) showed their distribution according to the treatments. The cluster represents the distance measure using the Euclidean algorithm and the clustering algorithm using ward.D. Hierarchical clustering was performed with the hclust function in the stat package of Microbiome-Analyst. Many filamentous fungi were observed in the heatmap, but their relative abundance was not homogeneous within each treatment. Some yeast species showed some association with the treatments (Appendix A). In particular, the control group was characterized by a high frequency of *Hanseniaspora*, *Peynorellaea*, *Cladosporium*, *Alternaria, Aureobasidium* genera, and unidentified fungi. These genera included plant pathogens and were prevalent in the grapes collected from plants treated with conventional agro-pharmaceutical (IPM). The samples R3P18 and R8P1 of the control treatment appeared similar but showed different species abundance with respect to the other three samples (R5P8, R6P11, R7P13) located in the central portion of the parcel control. The samples collected from row 9 (R9P3 e R9P11; Bion) showed a fungal composition similar to R5P8, R6P11, and R7P13 of the control group. A prevalence of fermenting yeasts belonging to the *Candida* genus was observed in the other three samples of the bion group (R9P11, R10P16, R11P7). The Water treatment was characterized by a sort of grouping, distinguishing between row 13 samples and row 14, where the first prevalence of the *Candida* genus was observed. 

Overall, the water samples were characterized by a low number of fungal species.

#### Signature

Linear Discriminant Analysis Effect Size (LDA-LEfSe) was applied at the species level to determine the metagenomic signature; it aimed to identify the fungal species with significant differential abundance according to the treatment.

In general, 66 OTUs have been identified and taxonomically assigned, as reported in Table 2.

The treatment parameter determined the different distributions of abundances of the fungal species. At the top of Table 2, the most relevant yeast species in the bion treatment are reported. As indicated by different colors, comparing the bion treatment with the control, it is possible to highlight a different species prevalence. In all the theses, the trend in terms of yeast number belonging to each species was similar but was higher in the bion treatment compared to the others. Table 2 reported *Hanseniaspora uvarum* as the dominant species in all the treatments. A substantial part of the total genome analyzed was unassigned at the species level and indicated as *incertae sedis*.

Yeast species such as *H. uvarum*, *Issatchenkia terricola* (or *Pichia terricola*), *Candida californica*, *Meyerozyma guilliermondii* (*basionym Pichia guilliermondii*), *C. stellata*, and *C. intermedia* were more abundant in the water treatment.

Considering the LEfSe analysis, it can be observed that some species were inhibited by the treatments. In fact, fourteen species, for example, *Aspergillus niger*, *Tricholoma saponaceum*, *Acremonium implicatum*, *Lophiostoma macrostomum*, and *Stemphylium herbarum,* have not been detected in the water treatment. In the bion treatment, *Saccharomycodes ludwigii*, *Scleroramularia abundans*, *Russula* sp., *Thelephoraceae* sp., and *Periconia* sp. were absent. Species such as *Mortierella horticola*, *Inocybe assimilata*, *Candida stellata*, *Candida intermedia*, *Candida stellimalicola*, *Sccharomycodes ludwigii*, and *Dissoconium* sp. did not occur in the control.

Specific fungal signatures were outlined by comparing the abundance of the Control and Bion treatments in order to verify if BION^®^50WG could affect the mycobiota.

Specific fungal signatures were outlined by comparing the abundance of the Control and Bion treatments in order to verify if BION^®^50WG could affect the mycobiota. Excluding *H*. *uvarum*, which was prevalent in each case, the bion treatment exhibited a relevant presence *Issatchenkia terricola*. *Fusarium* sp., *Oidiodendron* sp., *Saccharomycopsis crataegensis*, *Candida stellata*, *Candida intermedia*, *Candida stellimalicola*, *Metschnikowia pulcherrima*, *Zygosaccharomyces bisporus*, *Dissoconium* sp. from the *Ascomycota* phylum, as well as *Rozellomycota* phylum. Among the *Basidiomycota*, the prevalent species in the Bion group were *Cryptococcus chernovii*, *Bensingtonia* sp., *Udeniomyces pyricola*, *Inocybe assimilata*, *Tremellales* sp. Additionally, *Mortierella horticola* from the *Zygomycota* phylum was also prevalent in the Bion treatment.

In the Control group, the *Ascomycota* phylum was characterized by *Lophiostoma macrostomum*, *Rhodotorula* sp., *Ramichloridium indicum*, *Saccharomycetales* sp., *Dothideomycetes* sp., *Cenococcum geophilum*., and *Candida californica.* Among the *Basidiomycota* in the Control group, the prevalent species were *Atheliaceae* sp., *Hysterangium thwaitesii*, *Russula vesca*, *Russula* sp., *Agaricales* sp. and *Filobasidium floriforme*. *Mortierella pseudozygospora* was the most abundant species of *Zygomycota* phylum in Control.

A graphical overview of the relative abundance of the yeast species for each treatment can be observed in Appendix A.

## 4. Discussion

The present work aimed to describe the mycobiota associated with the grape bunches coming from grapevines subjected to three pest management conditions: integrated pest management (IPM), IPM combined with the plant biostimulant BION^®^50WG (Syngenta), and IPM combined with water nebulization.

The results highlighted that two-year treatment with BION^®^50WG did not determine significant differences in the biodiversity of the mycobiota, confirming that the geographical sampling site (climatic conditions, seasonal pattern, pollution, soil conditions, etc.) has a decisive impact on the microbial structure, as well as the fungal microbiota balance [16,69,70]. An analogous conclusion was achieved by Perazzoli and coworkers [71], which assumed that the phyllosphere microbiota of the grapevine was minimally affected by the tested treatments, although they mainly differed according to grapevine geographical localization. Only the alpha diversity relating to the “observed” species was statistically significant (*p*-value 0.04999) among the samples. It is noticed that the water nebulization produced a depletion in the composition of the yeast community of the berry skin, as demonstrated in the box plot in Figure 3. The highest number of fungal species observed was found in the bion samples and the lowest was recorded in the water theses, which was probably due to the washout effect of the water nebulization (Figure 3A and Appendix A). The commercial product BION^®^50WG is formulated as water-dispersible granules and used in water suspension by foliar treatment. Therefore, it is not comparable to the action of water or the co-formulate, which could have a certain adhesive action. Usually, plant protection products have chemical–physical properties that allow the active molecule to penetrate and not slip from the leaf. 

Considering Shannon’s and Simpson’s indexes, the alpha diversity was similar in all the considered treatments. Appendix A reported the value of all indexes for each sample, and it is possible to observe that the highest Shannon’s index value of the mycobiota of the berry skin was measured in samples R6P11 (C) (1.67), R9P11 (B) (1.59), and R10P16 (B) (1.61). On the contrary, the samples R3P18 (C) (1.14), R8P1 (C) (1.05), and R9P3 (B) (1.11) showed the lowest fungal biodiversity. The values of Shannon’s alpha diversity in the Control and Bion theses were scattered, while the water group was closest to the median (Figure 3B). The high Simpson’s index value (0.618) reflects a highly uneven community since the probability that two OTUs chosen at random are the same is high (Figure 3C).

The statistical analysis of the diversity among the samples shows that they are very similar in terms of the composition of fungal communities. In fact, the beta diversity was very similar in all the theses. We can suppose that the grouping of the samples is independent of the treatments but is probably affected by the climatic conditions and position, according to Vaudano and coworkers [50]. The confirmation of this hypothesis can be explained by looking at the heatmap graphic, which shows the composition of the fungal communities in terms of the main species and their respective abundance.

The similarity among the values of the alpha and beta diversity indexes is confirmed by the low difference of the species belonging to the signature, as described below.

The data processing such as a pie chart by Microbiome-Analyst (Appendix A) demonstrates that not-assigned OTUs represent a conspicuous group in terms of relative abundance. This is clearly a limit of the metabarcoding method since, for some reads, there are no corresponding entries in the reference ITS database, especially for *fungi* not yet cultivated or that are unculturable in laboratory conditions [71].

From the analysis of the core microbiota that focused on the *fungi* kingdom (Figure 5), is evident that *Hanseniaspora uvarum* was the dominant species in the three theses, in particular in the water treatment, confirming that this is the most common yeast found at ripening time [42]. In fact, *H. uvarum* is highly adapted to living on grapes thanks to the ability of the strains belonging to this species to metabolize some microbial toxins [72].

The heatmap reported in Figure 6 highlights a substantial uniformity among samples R3P18 (C), R8P1 (C), R13P6 (W), and R13P13 (W), which are characterized by a low abundance in yeasts. This result is probably due to their position close to the vineyard edge and more exposed to wind and a drought environment.

The bunches sampled in the center of the parcel control (R6P11 and R7P13) showed similar composition in terms of identified species and their abundance. In terms of fungi species and abundance, sample R5P8 (C) seems to behave as an intermediate between R3P18 (C) and R6P11 (C) or R7P13 (C). Samples collected from row 9 (B) have a greater similarity with the central samples of the control treatment, which is probably due to the drift effect, which did not allow a uniform distribution of the BION 50WG^®^. Thus, R9P3 (B) and R9P11 (B) show more similitude to R6P11 (C) or R7P13 (C) than the other three bunches collected from the bion parcel. R10P6 (B), R10P16 (B), and R11P7 (B) are quite similar, having in common only a high *Candida* sp. abundance. Thus, the biostimulant treatment seems to favor the development of higher biodiversity. *H. uvarum*, which is considered ubiquitous in the literature, is present in lower abundance in the bion treatment than in the control or water.

On the contrary, plants belonging to the water treatment demonstrate a low fungal biodiversity on the berry, which is probably due to the washout of the foliar irrigation that involves the bunches. The sample collected from row 13 showed the prevalence of the *Candida* genus, in particular the species *C. californica*. The samples from row 14 showed a high abundance of *Pleosporales* sp.*, Aureobasidium* sp., *H. uvarum,* and *C. diversa*. 

Concerning the signature, no significant differences can be observed; thus, it is impossible to identify a species as a biomarker associated with a specific treatment. However, as described above, we observed that some species were inhibited by the treatments.

Through the ratio between bion and control abundance (Table 2) and the proliferation of some yeasts, such as *Issatchenkia terricola*, *Saccharomycopsis crataegensis*, *Candida stellata*, *C. intermedia*, *C. stellimalicola*, *Zygosaccharomyces bisporus*, *Tremellales* sp., and *Dissoconium* sp. probably favored by BION^®^50WG, has been highlighted. 

Analyzing the mycobiota of montepulciano cv. in organic (OM), conventional (CM) and not-treated management (NTM), Agarbati and coworkers [43] did not find significant differences among the treatments and *H. uvarum,* which resulted in the predominant yeast in each sample. Similarly, our samples of Barbera cv. *H. uvarum* (43.96% in the control, 38.28% in bion, and 47.88% in water) were dominant in the grape environment, irrespective of the treatment. Its higher relative abundance in the water treatment is explained by the erosion of 15 species due to the washout of the irrigation. Regarding *Aureobasidium pullulans* (oxidative yeast-like), Agarbati and colleagues [43] detected this species in each condition with the highest prevalence in NMT (50%), underlining that the lack of treatment promotes its growth. Further studies revealed that this microorganism was one of the most abundant at harvest time [23,73]. On the other hand, in our study, we detected a low presence of *A. pullulans* in each treatment. The pie chart by Microbiome-Analyst (Appendix A) recorded just the bion group with 10% of abundance, which was not observed in the control and water groups.

Yeast species such as *H. uvarum*, *I. terricola*, *Meyerozyma guilliermondii*, and *Candida* sp., normally present on mature grapes and in initial phases of fermenting must [42,73] weremore abundant in the water treatment. It is possible to hypothesize that they take advantage of the high humidity or the lower presence of other species deriving from foliar irrigation. On the other hand, some species such as *Aspergillus niger*, *Tricholoma saponaceum*, *Udeniomyces pyricola*, *Acremonium implicatum*, and *Stemphylium herbarum* could be washed away from the bunches as previously assumed and are thus undetectable.

The composition of mycobiota described for each thesis is similar to that reported in the review by De Filippis and coworkers [74]. In fact, during the initial stages of fermentation, the high abundance of non-*Saccharomyces* yeast, such as *Hanseniaspora uvarum*, *Pichia kluyveri*, *Candida stellata*, *Metschnikowia pulcherrima*, and *I. terricola*, is typical in the berries’ skin [42].

## 5. Conclusions

The aim of this research was to evaluate the potential variation in the mycobiota due to biostimulant BION^®^50WG application in vineyards subjected to integrated pest management (IPM). We used a metabarcoding approach to compare the fungal communities on grapes from three experimental conditions: IPM, IPM+BION^®^50WG, and IPM+water foliar nebulization. We analyzed alpha and beta diversity, the microbial biodiversity index, and specific fungal signatures to assess the impact of the biostimulant on the yeast and filamentous fungi populations.

Based on the results obtained from our study, it can be concluded that the application of the biostimulant BION^®^50WG does not seem to significantly alter the mycobiota of grape bunches at ripening times. This finding suggests that the use of biostimulants as an alternative to traditional agrochemicals may not negatively impact the fungal communities on grape surfaces. Consequently, it is likely that the wine-making process will not be affected by the treatment with BION^®^, thus maintaining all the microbiological features characterizing the “terroir”. However, to strengthen our hypothesis, spontaneous or driven fermentation on BION^®^50WG-treated grapes should be carried on. These results are valuable for vineyard managers and farmers who seek alternatives to conventional agrochemicals while maintaining the health and quality of their crops. By exploring the potential effects of biostimulants on vineyard microbiota, we contribute to promoting sustainable practices in agriculture and the agri-food sector. However, it is essential to recognize that further studies may be necessary to understand the long-term effects of biostimulant applications on vineyard ecosystems and wine features fully.

In conclusion, our study sheds light on the potential of biostimulants as a sustainable option for plant protection in viticulture. The absence of significant differences in the mycobiota of grapevines subjected to the three treatments supports the notion that biostimulants can be considered promising alternatives to agrochemicals, contributing to the European Green Deal’s objectives and sustainable agriculture practices. Nonetheless, continuous monitoring and research in this field are crucial to ensure the safe and effective implementation of these innovative approaches in vineyard management.

## Figures and Tables

**Figure 1 microorganisms-11-01873-f001:**
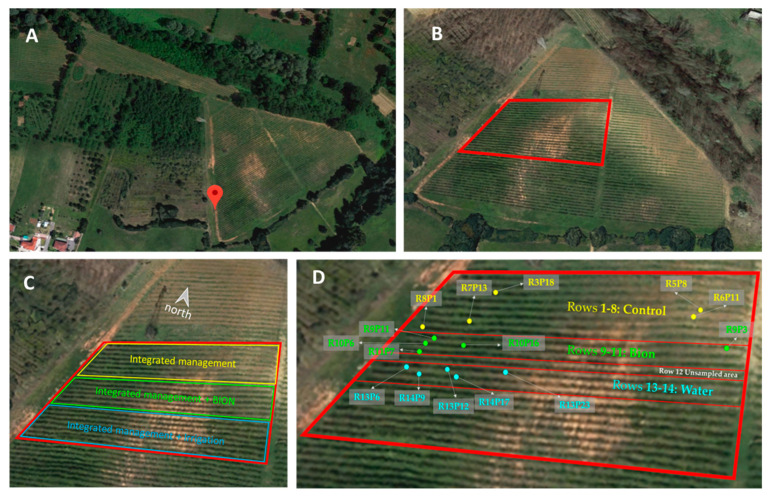
Sampling area details. (**A**) GPS picture of the vineyard area; (**B**) top view of the CREA—Viticoltura Enologia experimental vineyard. *Vitis vinifera* cv. Barbera was highlighted by a red line; (**C**) area of *V. vinifera* cv. Barbera with the indication of the three treatment areas; (**D**) sampling points.

**Figure 2 microorganisms-11-01873-f002:**
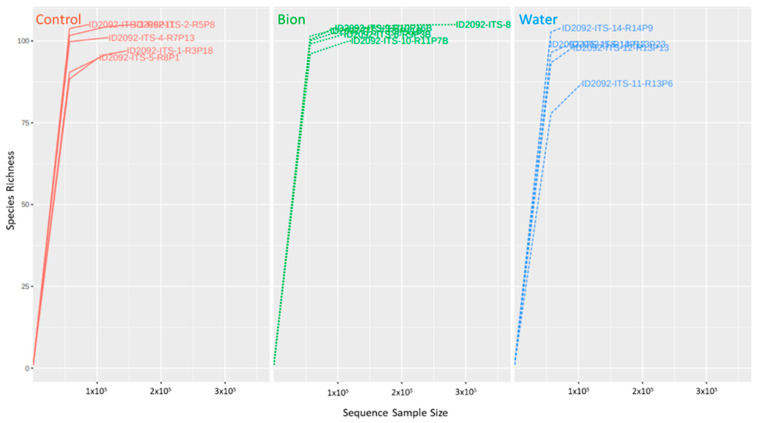
Rarefaction curves. Control group in red; Bion group in green; Water group in blue; x-axis: species richness or number of OTUs; y-axis: sequence sample size or number of reads.

**Figure 3 microorganisms-11-01873-f003:**
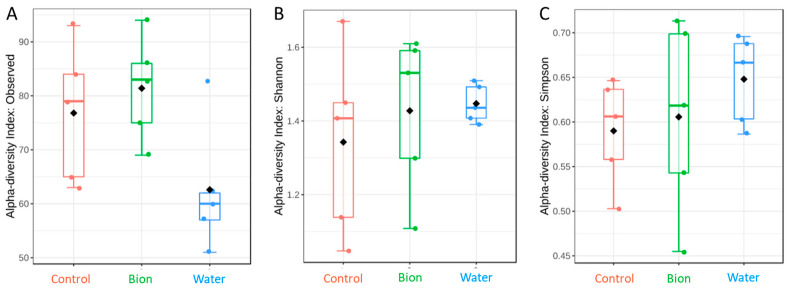
Alpha diversity. Alpha diversity analysis at the species level is estimated as the number of species observed (*p*-value = 0.04999) in (**A**), a Shannon’s index (*p*-value = 0.73345) in (**B**), as Simpson’s index (*p*-value = 0.56553) in (**C**). The *p*-value cut-off for significance is 0.05. The black dot indicates the mean value while the insides of the colored rectangles represent the median value. (In red, the Control; in green, Bion; in blue, Water.)

**Figure 4 microorganisms-11-01873-f004:**
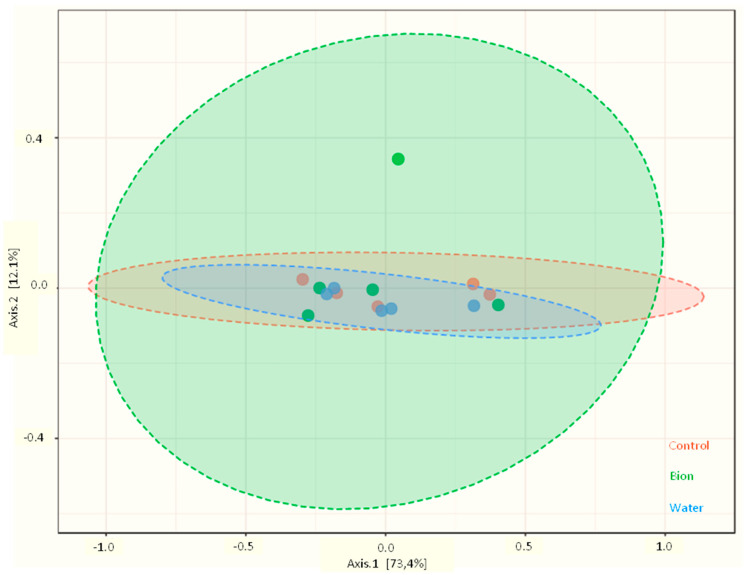
Beta diversity. PcoA based on Bray–Curtis metrics shows the dissimilarity of the fungal communities in the different samples according to the three theses (*p* = 0.688). (In red is the Control; in green is the Bion; in blue is the Water.)

**Figure 5 microorganisms-11-01873-f005:**
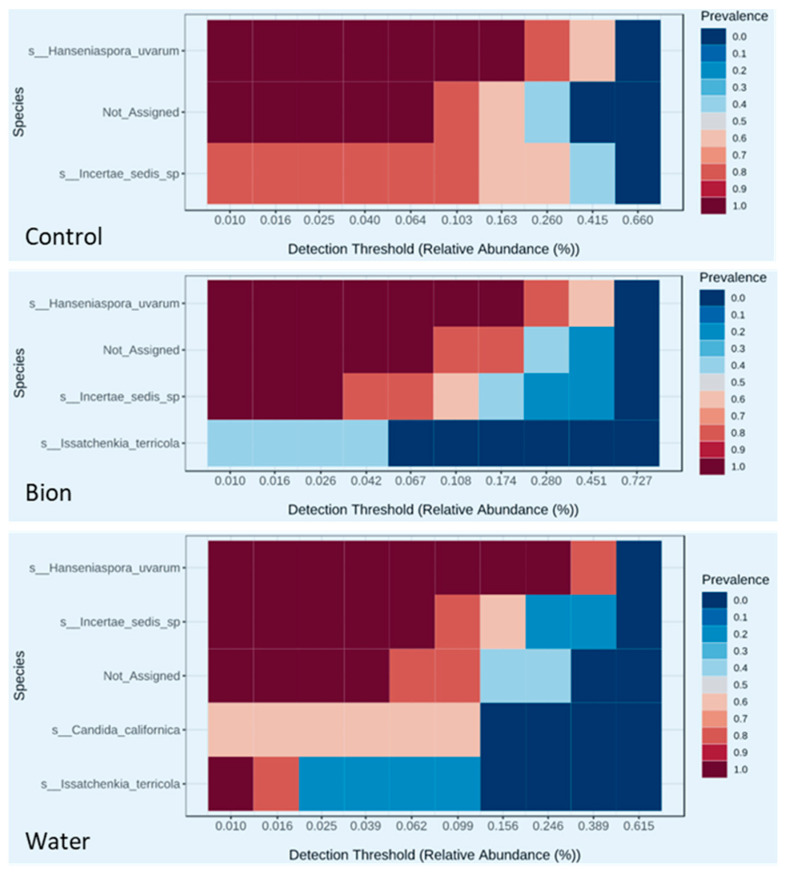
Core mycobiota. Representation of the main yeast at the species level for each treatment considering the median data from five samples for each treatment.

**Figure 6 microorganisms-11-01873-f006:**
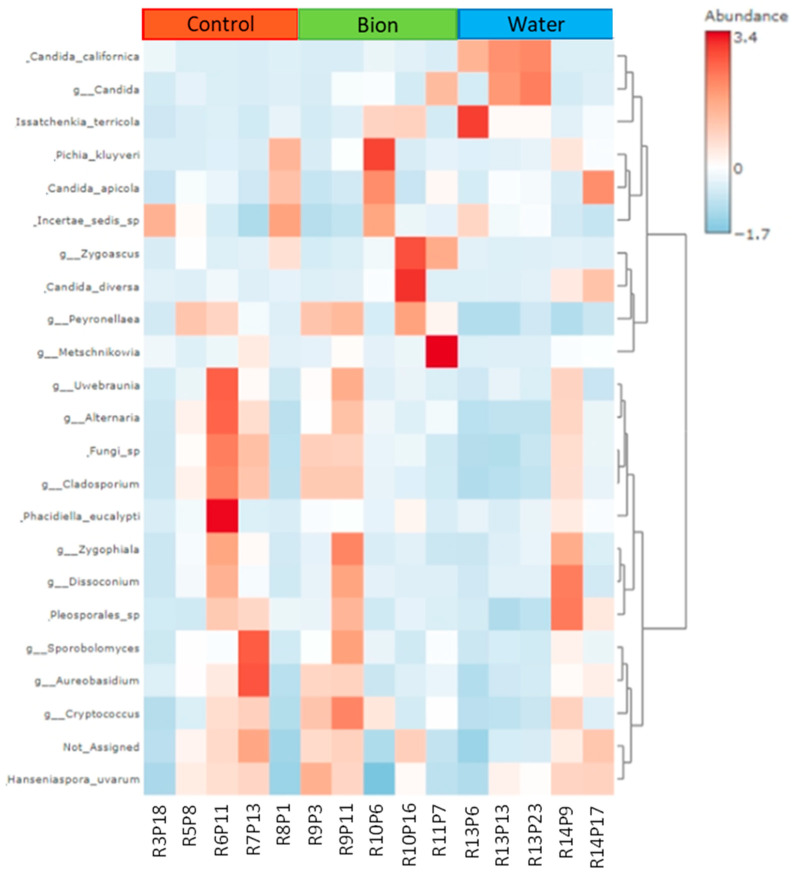
Heatmap. The graphical representation of the abundance at the species and higher taxa levels with respect to each sample (five per treatment). The resulting cluster shows the distance measure using Euclidean and clustering algorithms using ward.D. Hierarchical clustering is performed with the hclust function in the stat package of Microbiome-Analyst.

**Table 1 microorganisms-11-01873-t001:** Soil chemical composition of the vineyard.

Parameter	Unit of Measure	Results
Granulometry		
Total sand	%	31.25
Total silt	%	38.75
Clay	%	30.00
Analyzed compound		
pH (in water)	-	7.43
Total limestone (CaCO_3_)	%	0.41
Organic matter	%	1.50
Organic carbon	%	0.87
Total nitrogen	%	0.0706
C/N ratio	-	12.32
Absorbable phosphorus	ppm	10
Exchangeable potassium	ppm	59
Exchangeable calcium	ppm	2362
Exchangeable magnesium	ppm	245
CEC	meq/100 g	27.91

**Table 2 microorganisms-11-01873-t002:** LEfSe. Linear discriminant analysis at species level according to treatments (comprehensive of five samples per treatment). Light gray indicates species less present in the Bion treatment than the control, and red indicates a presence of less than 50%. Yellow indicates that the species is more present in the Bion than the control, and green indicates a presence of more than 200%. LEfSe results using the non-parametric factorial Kruskal–Wallis (KW) sum-rank test. Adjusted *p*-value cut-off = 0.05 and LDA score = 1.0.

Species	Phylum *	*p*-Values	Control	BION	Water	LDA Score	BION/Control (%)
*Hanseniaspora uvarum*	*Ascomycota*	0.811	4,428,800.00	4,599,000.00	5,000,900.00	5.46	103.84
*Not Assigned*		0.566	2,073,400.00	2,819,500.00	1,607,200.00	5.78	135.98
*Issatchenkia terricola*	*Ascomycota*	0.065	56,298.00	216,560.00	374,330.00	5.2	384.67
*Fusarium* sp.	*Ascomycota*	0.579	35,122.00	92,297.00	26,681.00	1.53	262.79
*Oidiodendron* sp.	*Ascomycota*	0.731	22,559.00	83,656.00	46.34	1.5	370.83
*Aspergillus niger*	*Ascomycota*	0.266	52,233.00	69,658.00	0.00	1.43	133.36
*Rozellomycota* sp.	*Rozellomycota*	0.980	33,436.00	69,658.00	21,856.00	1.15	208.33
*Saccharomycopsis crataegensis*	*Ascomycota*	0.594	13,584.00	69,387.00	72,319.00	1.48	510.80
*Tricholoma saponaceum*	*Basidiomycota*	0.581	54,336.00	61,881.00	0.00	1.5	113.89
*Bensingtonia* sp.	*Basidiomycota*	0.266	12,563.00	38,813.00	0.00	1.31	308.95
*Udeniomyces pyricola*	*Basidiomycota*	0.581	16,718.00	36,133.00	0.00	1.28	216.13
*Mortierella horticola*	*Zygomycota*	0.581	0.00	36,133.00	26,681.00	1.28	∞
*Acremonium implicatum*	*Ascomycota*	0.581	22,559.00	36,133.00	0.00	1.28	160.17
*Inocybe assimilata*	*Basidiomycota*	0.117	0.00	31,998.00	0.00	1.23	∞
*Hansfordia pulvinata*	*Ascomycota*	0.980	19,325.00	27,863.00	26,681.00	0.722	144.18
*Candida diversa*	*Ascomycota*	0.691	2784.70	21,917.00	14,604.00	3.98	787.05
*Candida apicola*	*Ascomycota*	0.691	17,703.00	21,485.00	25,185.00	3.57	121.36
*Candida stellata*	*Ascomycota*	0.266	0.00	13,932.00	45,026.00	1.37	∞
*Metschnikowia pulcherrima*	*Ascomycota*	0.049	2744.50	8929.20	406.45	3.63	325.35
*Pichia kluyveri*	*Ascomycota*	0.566	4297.40	8020.60	3915.90	3.31	186.64
*Cryptococcus chernovii*	*Basidiomycota*	0.045	524.73	1089.10	139.51	2.68	207.55
*Zygosaccharomyces bisporus*	*Ascomycota*	0.925	169.14	1010.10	285.43	2.62	597.20
*Articulospora* sp.	*Ascomycota*	0.681	624.23	813.06	132.49	2.53	130.25
*Cryptococcus* sp.	*Basidiomycota*	0.098	446.47	748.62	213.42	2.43	167.68
*Zygosaccharomyces bailii*	*Ascomycota*	0.444	391.04	661.75	553.30	2.13	169.23
*Alternaria alternata*	*Ascomycota*	0.160	429.50	501.99	248.13	2.11	116.88
*Paraconiothyrium hawaiiense*	*Ascomycota*	0.697	306.65	422.43	915.95	2.49	137.76
*Exobasidiomycetes* sp.	*Basidiomycota*	0.930	290.07	417.58	152.33	2.13	143.96
*Candida intermedia*	*Ascomycota*	0.101	0.00	403.00	26,681.00	2.31	∞
*Candida stellimalicola*	*Ascomycota*	0.117	0.00	386.02	0.00	2.29	∞
*Sporobolomyces coprosmae*	*Basidiomycota*	0.163	310.06	379.60	23.17	2.25	122.43
*Tremellales* sp.	*Basidiomycota*	0.309	47.02	261.48	77,112.00	2.03	556.10
*Dissoconium* sp.	*Ascomycota*	0.130	0.00	104.55	53,362.00	1.73	∞
*Saccharomycodes ludwigii*	*Ascomycota*	0.032	0.00	0.00	540.83	2.43	-
*Cenococcum geophilum*	*Ascomycota*	0.225	49,627.00	43,099.00	105.11	1.51	86.85
*Atheliaceae* sp.	*Basidiomycota*	0.581	81,505.00	41,254.00	0.00	1.62	50.62
*Candida californica*	*Ascomycota*	0.114	44,246.00	38,470.00	776,360.00	5.57	86.95
*Hysterangium thwaitesii*	*Basidiomycota*	0.581	62,815.00	32,784.00	0.00	1.51	52.19
*Rhodotorula* sp.	*Ascomycota*	0.772	70,243.00	31,998.00	36,298.00	1.3	45.55
*Mortierella pseudozygospora*	*Zygomycota*	0.891	33,436.00	27,593.00	23.17	0.788	82.52
*Lophiostoma macrostomum*	*Ascomycota*	0.077	81,121.00	22,422.00	0.00	1.62	27.64
*Agaricales* sp.	*Basidiomycota*	0.581	45,117.00	20,627.00	0.00	1.37	45.72
*Fungi* sp.		0.230	23,879.00	20,067.00	11,400.00	3.8	84.04
*Tomentella* sp.	*Basidiomycota*	0.544	22,559.00	18,067.00	49,851.00	1.23	80.09
*Meyerozyma guilliermondii*	*Ascomycota*	0.544	19,325.00	1327.90	53,362.00	2.82	6.87
*Pleosporales* sp.	*Ascomycota*	0.733	1287.80	1218.50	1104.70	1.97	94.62
*Saccharomycetales* sp.	*Ascomycota*	0.471	54,446.00	768.17	341.24	2.55	1.41
*Phacidiella eucalypti*	*Ascomycota*	0.742	1370.50	591.07	565.06	2.61	43.13
*Chalastospora ellipsoidea*	*Ascomycota*	0.465	676.11	384.99	286.01	2.29	56.94
*Stemphylium herbarum*	*Ascomycota*	0.017	307.83	303.24	0.00	2.19	98.51
*Ramichloridium indicum*	*Ascomycota*	0.859	55,994.00	255.30	266.81	2.03	0.46
*Mycosphaerellaceae* sp.	*Ascomycota*	0.826	357.27	230.47	318.80	1.81	64.51
*Filobasidium floriforme*	*Basidiomycota*	0.110	58,562.00	213.77	23.17	1.98	0.37
*Aureobasidium pullulans*	*Ascomycota*	0.821	124.55	183.40	122.45	1.5	147.25
*Acaromyces ingoldii*	*Basidiomycota*	0.455	842.59	168.87	286.04	2.53	20.04
*Tilletiopsis pallescens*	*Basidiomycota*	0.288	291.83	165.72	53,362.00	2.08	56.79
*Ascomycota* sp.	*Ascomycota*	0.426	55,994.00	155.92	36,298.00	1.78	0.28
*Cryptococcus victoriae*	*Basidiomycota*	0.057	137.25	136.26	0.00	1.84	99.28
*Russula vesca*	*Basidiomycota*	0.631	66,872.00	132.13	26,681.00	1.73	0.20
*Dothideomycetes* sp.	*Ascomycota*	0.421	50,252.00	113.04	26,681.00	1.65	0.22
*Tremellomycetes* sp.	*Basidiomycota*	0.199	22,559.00	103.60	26,681.00	1.62	0.46
*Devriesia pseudoamericana*	*Ascomycota*	0.631	22,559.00	65.47	18,958.00	1.38	0.29
*Russula* sp.	*Basidiomycota*	0.117	84,394.00	0.00	0.00	1.64	0
*Thelephoraceae* sp.	*Basidiomycota*	0.266	16,718.00	0.00	59,468.00	1.49	0.00
*Scleroramularia abundans*	*Ascomycota*	0.581	16,718.00	0.00	53,362.00	1.44	0.00
*Periconia* sp.	*Ascomycota*	0.581	45,117.00	0.00	26,681.00	1.37	0.00
*Incertae sedis* sp.		0.698	3,335,700.00	2,231,500.00	2,178,100.00	5.76	66.90

* The Mycobank database, https://www.mycobank.org/Simple%20names%20search (assessed on 24 April 2023) was used to associate the species at the corresponding phylum.

## Data Availability

The genomic sequences were included in the BioProject PRJNA801453 “Study of the yeast microbiota of grape skin” available in NCBI database accessed on 28 January 2022.

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
