# Peer review of "The Impact of a Commercial Biostimulant on the Grape Mycobiota of Vitis vinifera cv. Barbera"

_microorganisms, 2023, doi:10.3390/microorganisms11081873_

Round 1

Reviewer 1 Report

This work discussed the effect of a biostimulant on the microbiota on grape skin. Sufficient background (climate, soil and vineyard management) was introduced and the microbiota profiles were investigated. The topic is interesting with some novel results; however, some major flaws exist and weaken the content. The writing of the MS needs to be largely improved, and plenty of minor errors exist in its current version. I only provide a non-exhaustive list as below:

L7: country information should be added in the address.

L16: Too much background was described and more specific results were encouraged to supply in the abstract.

L141, the “V. Vinifera” should be italic.

Line 306, “measures a soil’s ability to …”

L334, “Alfa” should be corrected to alpha. And the alpha diversity should be calculated at otu level.

Line 366, change to ‘did not’

L368, The method of significance testing needs to be mentioned.

Line 375, “the other three samples”?

Line 382, “the Water theses”? could not understand!

L394, please omit the “Effect size”.

L398, p in the “p value” should be italic.

Besides, as you mentioned in the introduction, the biostimulant would regulate the microbiota by activating the host immunity system, so the gene expression profile related to plant immunity should be determined or at least discussed.

Line 540, what is the “Water thesis”?, and what is meaning of ‘thesis ’ in line 542, do you mean ‘treatments’? The conclusion needs to be rewritten.

 Extensive editing of English language required

Author Response

Dear reviewer,

below are our observations in green:

Open Review

Quality of English Language

( ) I am not qualified to assess the quality of English in this paper
( ) English very difficult to understand/incomprehensible
(x) Extensive editing of English language required
( ) Moderate editing of English language required
( ) Minor editing of English language required
( ) English language fine. No issues detected

Yes

Can be improved

Must be improved

Not applicable

Does the introduction provide sufficient background and include all relevant references?

( )

(x)

( )

( )

Lines 16-18 have been cancelled

The sentence (Line 21-23) has been changed to:

“Their application could potentially open new scenarios in the composition of the vineyard microbiota (grapevine fitness, productivity, etc.), and of the grapes’ microflora on the grape surface (grape quality), in the perspective that could elicit consequences in wine features.”

The last introduction paragraph (Line 31-34) has been changed to:

“…comparing the abundance of yeast and filamentous fungi in IPM and BION®50WG treatments. From the biodiversity data analysed no significant differences was detected.”

Are all the cited references relevant to the research?

(x)

( )

( )

( )

We added some other references

38. PubChem website, Acibenzolar-S-Methyl compound. Available online: https://pubchem.ncbi.nlm.nih.gov/compound/86412 (accessed on 7 July 2022)

58. Bona, E.; Massa, N.; Toumatia, O.; Novello, G.; Cesaro, P.; Todeschini, V.; Boatti, L.; Mignone, F.; Titouah, H.; Zitouni, A.; et al. Climatic Zone and Soil Properties Determine the Biodiversity of the Soil Bacterial Communities Associated to Native Plants from Desert Areas of North-Central Algeria. Microorganisms 2021, 9, 1359, doi:10.3390/microorganisms9071359

Is the research design appropriate?

(x)

( )

( )

( )

Are the methods adequately described?

( )

(x)

( )

( )

We  improved the methods.

Are the results clearly presented?

( )

( )

(x)

( )

We tried to improve the results (Line 377-378), also with supplementary figures.

Are the conclusions supported by the results?

( )

(x)

( )

( )

We changed most of the conclusions

Line 548-579

Comments and Suggestions for Authors

This work discussed the effect of a biostimulant on the microbiota on grape skin. Sufficient background (climate, soil and vineyard management) was introduced and the microbiota profiles were investigated. The topic is interesting with some novel results; however, some major flaws exist and weaken the content. The writing of the MS needs to be largely improved, and plenty of minor errors exist in its current version. I only provide a non-exhaustive list as below:

L7: country information should be added in the address.

corrected

L16: Too much background was described and more specific results were encouraged to supply in the abstract. The abstract has been changed according to the suggestion.

L141, the “V. Vinifera” should be italic. It has been corrected.

Line 306, “measures a soil’s ability to …”.  Corrected

L334, “Alfa” should be corrected to alpha. And the alpha diversity should be calculated at otu level.

The initial data analysis had been made on OTUs identifications, subsequently using Microbiome-Analyst our attention was focused on the species.

Line 366, change to ‘did not’. Done

L368, The method of significance testing needs to be mentioned.

We add the sentence (Line 384-386: “The cluster represent the distance measure using Euclidean algorithm and clustering al-gorithm using ward.D. The hierarchical clustering is performed with the hclust function in stat package of Microbiome-Analyst.”

Line 375, “the other three samples”? We specified “(R5P8, R6P11, R7P13)”

Line 382, “the Water theses”? could not understand! The word ‘theses’ has been changed to ‘samples’

L394, please omit the “Effect size”.  Omitted

L398, p in the “p value” should be italic. Corrected

Besides, as you mentioned in the introduction, the biostimulant would regulate the microbiota by activating the host immunity system, so the gene expression profile related to plant immunity should be determined or at least discussed.

In our field of study, we are more focused on the microbial biodiversity in the vineyard and on the microorganisms’ role in oenology.

The study of SAR gene expression in grapevine is very interesting, especially to determine the efficacy of ASM in grapevine and its applicability. This is of particular interest for research groups involved in vine defence or in plant immunity by physiological response to synthetic biostimulants.

Line 540, what is the “Water thesis”?, and what is meaning of ‘thesis ’ in line 542, do you mean ‘treatments’? The word “thesis” has been changed to ‘treatment’

The conclusion needs to be rewritten.

We changed most of the conclusions paragraph.

best regards

Reviewer 2 Report

General comments

This manuscript shows us some interesting data about the effects of a commercial biostimulant "BION®50WG" on grape mycobiota of Vitis vinifera cv. Barbera. The manuscript is of good quality and the format of figures are standard. The methods are appropriate and properly conducted. To understand more about this research I read related references, and from my own view, this manuscript could be accepted for publication after a minor revision of the abstract and conclusion part and asittions of explanation about Introduction and Results on the effects on bacteria and other filamentous fungi. 

- Could you add further sentence of results on the effects on mycobiota (bacteria, yeast, and fungi) in Abstract section.

- As mentioned as Line94, ASM can prevent bacteriosis. Why does BION®50WG have antifungal activity against fungi? Could you explain or discuss these.

Author Response

Dear reviewer,

below are our observations in green:

Open Review

Quality of English Language

( ) I am not qualified to assess the quality of English in this paper
( ) English very difficult to understand/incomprehensible
( ) Extensive editing of English language required
( ) Moderate editing of English language required
( ) Minor editing of English language required
(x) English language fine. No issues detected

Yes

Can be improved

Must be improved

Not applicable

Does the introduction provide sufficient background and include all relevant references?

( )

(x)

( )

( )

Lines 16-18 have been cancelled

The sentence (Line 21-23) has been changed to:

“Their application could potentially open new scenarios in the composition of the vineyard microbiota (grapevine fitness, productivity, etc.), and of the grapes’ microflora on the grape surface (grape quality), in the perspective that could elicit consequences in wine features.”

The last introduction paragraph (Line 31-34) has been changed to:

“…comparing the abundance of yeast and filamentous fungi in IPM and BION®50WG treatments. From the biodiversity data analysed no significant differences was detected.”

Are all the cited references relevant to the research?

( )

(x)

( )

( )

We added some other references

38. PubChem website, Acibenzolar-S-Methyl compound. Available online: https://pubchem.ncbi.nlm.nih.gov/compound/86412 (accessed on 7 July 2022)

58. Bona, E.; Massa, N.; Toumatia, O.; Novello, G.; Cesaro, P.; Todeschini, V.; Boatti, L.; Mignone, F.; Titouah, H.; Zitouni, A.; et al. Climatic Zone and Soil Properties Determine the Biodiversity of the Soil Bacterial Communities Associated to Native Plants from Desert Areas of North-Central Algeria. Microorganisms 2021, 9, 1359, doi:10.3390/microorganisms9071359

Is the research design appropriate?

(x)

( )

( )

( )

Are the methods adequately described?

(x)

( )

( )

( )

Are the results clearly presented?

(x)

( )

( )

( )

We tried to improve the results (Line 377-378), also with supplementary figures.

Are the conclusions supported by the results?

( )

(x)

( )

( )

We changed most of the conclusions

Line 548-579

Comments and Suggestions for Authors

General comments

This manuscript shows us some interesting data about the effects of a commercial biostimulant "BION®50WG" on grape mycobiota of Vitis vinifera cv. Barbera. The manuscript is of good quality and the format of figures are standard. The methods are appropriate and properly conducted. To understand more about this research I read related references, and from my own view, this manuscript could be accepted for publication after a minor revision of the abstract and conclusion part and asittions of explanation about Introduction and Results on the effects on bacteria and other filamentous fungi.

Answer: In this work the metabarcoding analysis was conducted only on the fungal fraction of the microbiota, using the primers ITS1F and ITS4R strictly related to fungal species. Since the metabarcoding analysis on 16S related to prokaryotic organisms was not realised.

changed sentences with greater emphasis on filamentous fungi:

L.31-33: “The statistical analysis applied on the biodiversity indexes of fungal communities, both yeasts and filamentous fungi, showed no significant differences between the treatments.”

  1. 387-388: “Many filamentous fungi were relieved in the heatmap, but their relative abundance wasn’t homogeneous within each treatment.”
  2. 553-555: “From the LEfSe table appears as the filamentous fungi are fewer or absent in Bion and Water treatments, while the yeasts seem more inhibited in the Control treatment.”

- Could you add further sentence of results on the effects on mycobiota (bacteria, yeast, and fungi) in Abstract section.

We have changed the final sentence (Line 30-31) in the abstract: “By a metagenomic approach we revealed the magnitude of α and β diversity, the microbial biodiversity index and Specific fungal signatures were highlighted by comparing the abundance of yeast and filamentous fungi in IPM and BION®50WG treatments."

In this study we analysed only the fungal fraction of the microbiota (=mycobiota) because the different composition of the yeast communities, depending on the dominant species, could affect the alcoholic fermentation, especially the spontaneous one, and have a role in the development of the wine organoleptic features.

- As mentioned as Line94, ASM can prevent bacteriosis. Why does BION®50WG have antifungal activity against fungi? Could you explain or discuss these.

In order to explain the effect of BION50WG we added the sentence at Line 97-103: “From the label of the commercial product, BION®50WG is effective against bacterial diseases of some fruit plants and against the downy mildew of tobacco (Peronospora tabacina). Moreover, ASM is known as BTH (benzo (1,2,3) thiadia-zole-7-carbothioic acid S-methyl ester) and was used as fungicide on a variety of crop, for example in the control of the powdery mildew of wheat and barley [37,38]. ASM is classified as plant activator, antifungal agrochemical and profungicide (the corresponding carboxylic acid is released by hydrolysis of the thioester group) [38].”

Best regards

Round 2

Reviewer 1 Report

The authors have revised the MS to answer most of my techinique concerns.

 Extensive editing of English language required

Author Response

Dear Reviewer,

as suggested, English language has been edited.

Best regard